# Tardigrade secretory proteins protect biological structures from desiccation
Samuel Lim[1,2], Charles B. Reilly[2], Zeina Barghouti [3], Benedetto Marelli [4], Jeffrey C. Way[1,2] & Pamela A. Silver [1,2] ✉

Tardigrades, microscopic animals that survive a broad range of environmental stresses, express a unique set of proteins termed tardigrade-specific intrinsically disordered proteins (TDPs). TDPs are often expressed at high levels in tardigrades upon desiccation, and appear to mediate stress adaptation. Here, we focus on the proteins belonging to the secreted family of tardigrade proteins termed secretory-abundant heat soluble ("SAHS") proteins, and investigate their ability to protect diverse biological structures. Recombinantly expressed SAHS proteins prevent desiccated liposomes from fusion, and enhance desiccation tolerance of *E. coli* and *Rhizobium tropici* upon extracellular application. Molecular dynamics simulation and comparative structural analysis suggest a model by which SAHS proteins may undergo a structural transition upon desiccation, in which removal of water and solutes from a large internal cavity in SAHS proteins destabilizes the beta-sheet structure. These results highlight the potential application of SAHS proteins as stabilizing molecules for preservation of cells.

Water is the universal solvent essential for survival. However, several organisms demonstrate remarkable abilities to withstand an almost complete loss of water for a prolonged time and recover upon rehydration[1,2]. This phenomenon of "anhydrobiosis", or "life without water" has attracted a great deal of attention because it may reveal Nature's strategies to endure adverse environments and open up ways to achieve efficient preservation of living matter. However, the precise mechanisms behind anhydrobiosis remain elusive, as they may involve a complex combination of protective proteins, solutes, and cellular machinery working in a concert[3].

Several molecules have been identified as mediators of desiccation tolerance in various organisms. One is the disaccharide trehalose, which accumulates in a large quantity in animals such as brine shrimps or nematodes[4]. Trehalose uptake enhances the survival of yeast and human cells under drying conditions and may protect dehydrated cells by vitrification (turning the cell interior into a glass-like state)[4–6]. Several "intrinsically disordered proteins (IDPs)", which lack fixed tertiary structure, have also been associated with desiccation tolerance[2]. For instance, late embryogenesis-abundant (LEA) proteins are tolerance molecules found in plant seeds and anhydrobiotic species[7]. LEA proteins are mostly unstructured in an aqueous environment and assume helical structure upon desiccation to alleviate damage due to drying[7,8].

Recent studies on the survival mode of tardigrades revealed novel classes of stress proteins. Tardigrades (water bears) are microscopic animals that can survive under a broad range of environmental stressors, including desiccation, high and low temperatures, radiation, and even exposure to outer space[9–11]. Tardigrades express LEA proteins, and also three different families termed "tardigrade disordered proteins" (TDPs) unique to the phylum Tardigrada[12]. Some of these TDPs are highly disordered and less likely to undergo aggregation at high temperatures. (The classification of TDPs as disordered proteins is based on a sequence-based prediction[13].) Three different groups of TDPs were termed cytoplasmic-, mitochondrial- and secretory-abundant heat soluble (CAHS, MAHS, SAHS) proteins according to their putative subcellular localization (Fig. 1a). In each family, the proteins show sequence similarity[14,15]. TDPs are either constitutively expressed at high levels or enriched upon desiccation and knocking out even one CAHS or SAHS gene can cause diminished desiccation tolerance[13].

SAHS proteins are classified as secreted as they are generally encoded with signal sequences and are exported to the extracellular space upon expression in eukaryotes[14]. Sequence-based structural prediction suggested that, unlike CAHS and MAHS proteins that are estimated to be highly disordered, the 42 putative SAHS proteins from tardigrade species *Hypsibius exemplaris, Ramazzottius varieornatus, Paramacrobiotus richtersi* and *Milnesium tardigradum* are more ordered[13]. Fukuda et al. solved the structure of *R. varieornatus* SAHS1 and SAHS4[16,17], experimentally verifying that, at least under hydrated conditions, the SAHS proteins have defined

[1]Department of Systems Biology, Harvard Medical School, Boston, MA 02115, USA. [2]Wyss Institute for Biologically Inspired Engineering at Harvard University, Boston, MA 02115, USA. [3]Department of Mechanical Engineering, Massachusetts Institute of Technology, Cambridge, MA 02139, USA. [4]Department of Civil and Environmental Engineering, Massachusetts Institute of Technology, Cambridge, MA 02139, USA. ✉e-mail: pamela_silver@hms.harvard.edu

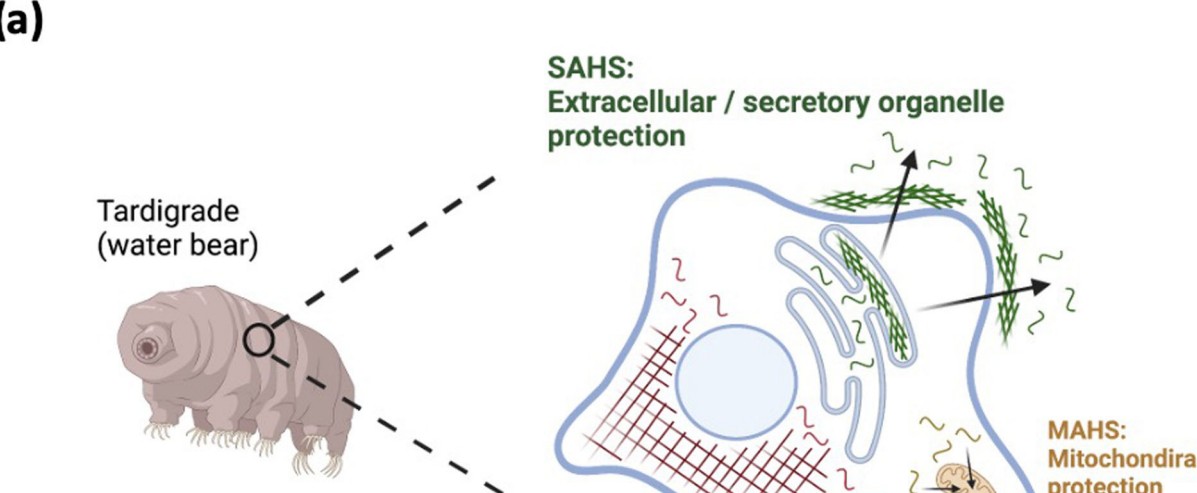

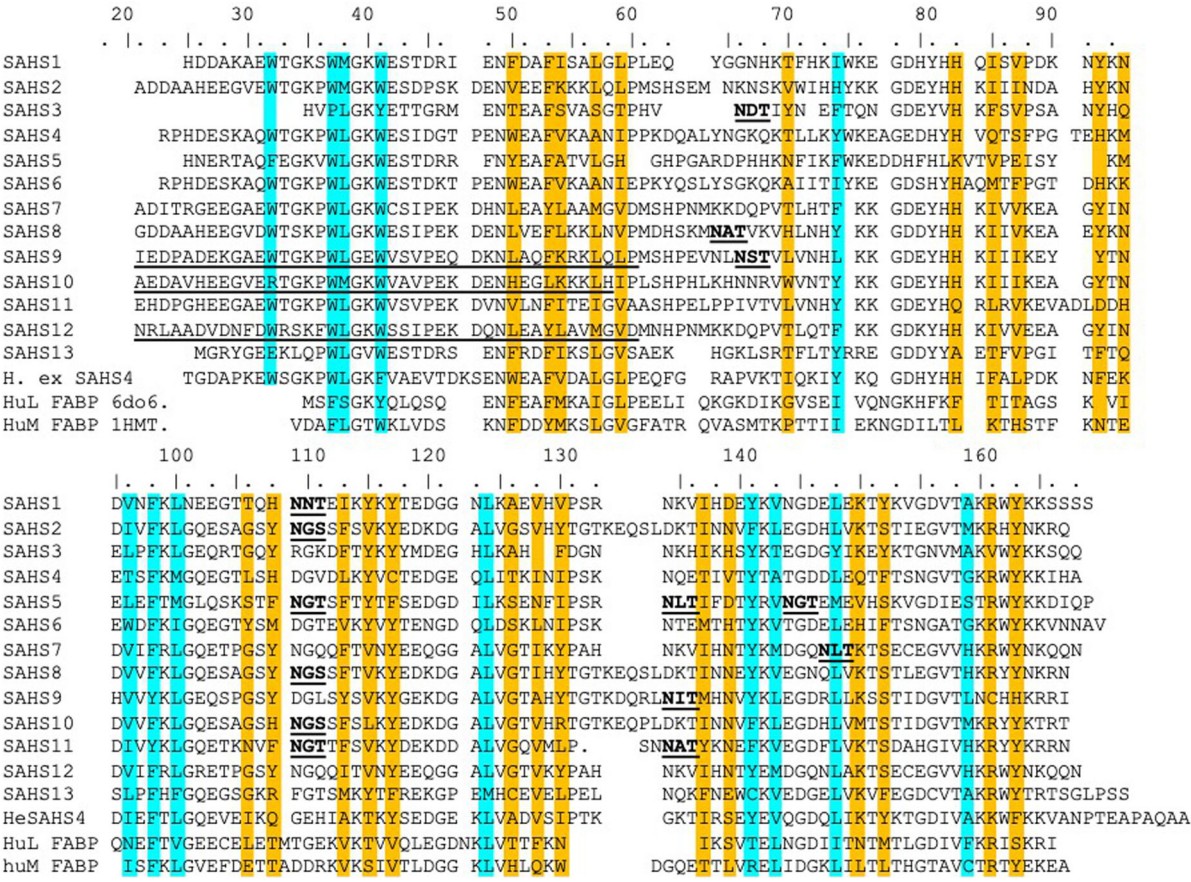

**Fig. 1 | Classification and sequence analysis of tardigrade SAHS proteins.**
**a** Classification of tardigrade-specific desiccation-protective proteins. Tardigrades express unique sets of proteins termed tardigrade-specific intrinsically disordered proteins (TDPs). TDPs are further classified into three different subgroups, Cytosolic-, Mitochondrial- and Secretory-abundant heat soluble (C/M/SAHS) proteins, based on their subcellular localization. Created with BioRender.com. **b** An alignment of SAHS1-13 of *R. varieornatus*, SAHS4 of *H. exemplaris*, and two FABP (fatty acid binding protein) sequences, with amino acids whose side chains point inward toward the open cavity highlighted in orange, and those with side chains in the hydrophobic core highlighted in cyan. Underlined amino acid sequences at the N-termini of SAHS9, 10 and 12 are newly identified based on analysis of the *R. varieornatus* genome sequence. Underlined and bold sequences are putative N-linked glycosylation sites based on the consensus sequence Asn-X-Ser/Thr (X not Pro).

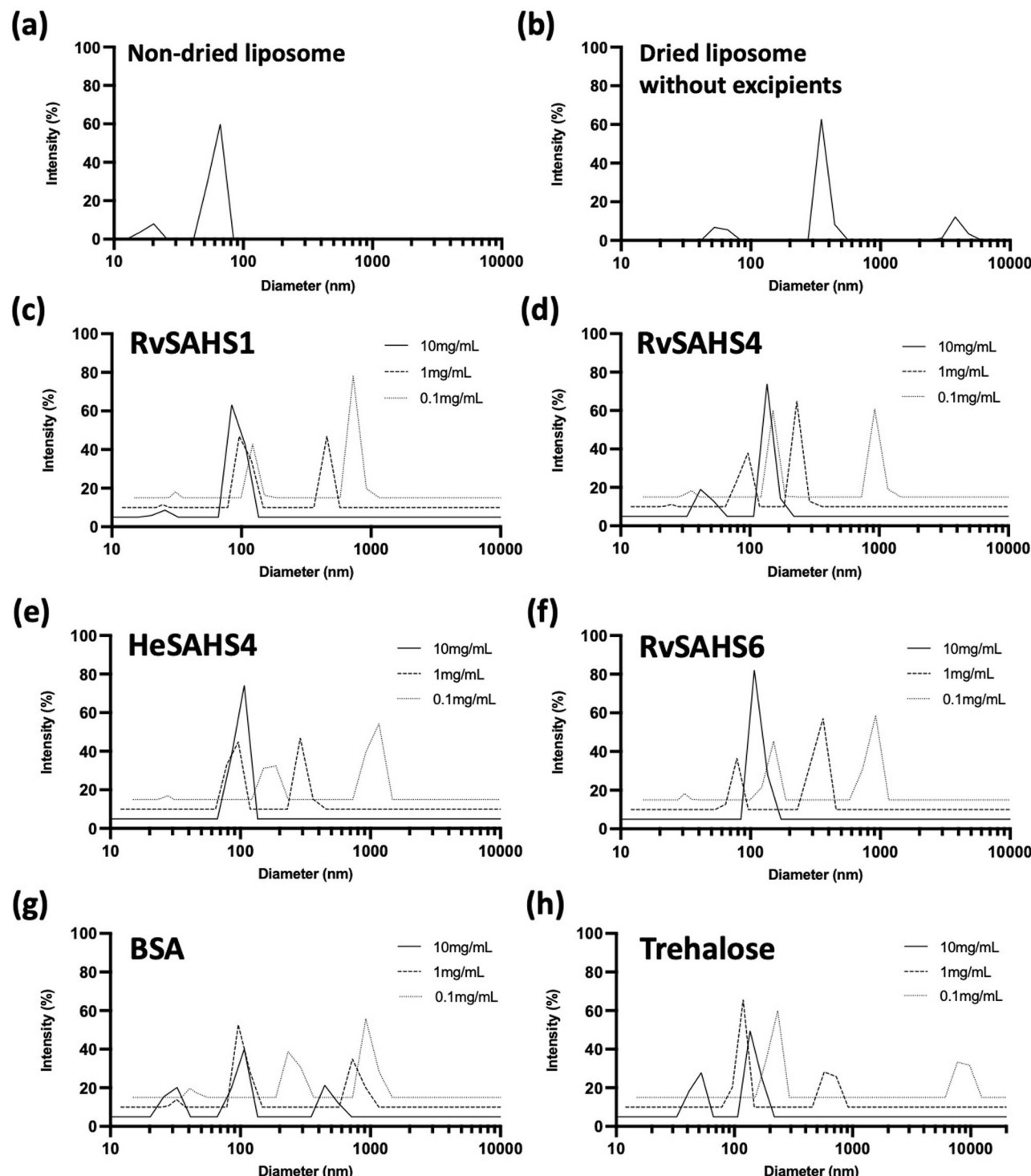

**Fig. 2 | Tardigrade SAHS proteins stabilize liposomes from desiccation-induced damage.** POPC liposomes at 1.4 mg/mL were dried with and without the addition of SAHS proteins and BSA at varying concentrations of 0.1–10 mg/mL, and their size distributions were measured by DLS. **a** Size distribution of non-dried POPC liposomes. **b** Size distribution of POPC liposomes dried and rehydrated without additives. **c–h** Size distributions of the liposomes dried with **c** RvSAHS1, **d** RvSAHS4, **e** HeSAHS4, **f** RvSAHS6, **g** BSA, and **h** trehalose.

structures, and their description as "intrinsically disordered proteins" is not accurate. RvSAHS1 and 4 proteins share sequence and structural similarity to fatty acid binding proteins (FABPs): their structures include β-barrels with large internal cavities that correspond to the fatty acid binding pockets in FABPs[16,17]. The function of these large cavities in the SAHS proteins is unclear, as is their biochemical mechanism in protecting against desiccation. Additionally, AlphaFold structures are available on the protein structure

database (PDB) for all SAHS proteins identified from *R. varieornatus* and *H. exemplaris*.

In this study, we investigated SAHS proteins from tardigrades *R. varieornatus* and *Hypsibius exemplaris* for their potential to protect various biological structures from desiccation-induced damage. We expressed four different SAHS proteins and found that they can specifically protect liposomes and bacterial cells but not enzymes upon desiccation. SAHS proteins

undergo structural changes under dehydration with trifluoroethanol. Finally, we performed molecular dynamic and bioinformatic analyses to identify structural features of the SAHS proteins that may contribute to desiccation protection. These studies lead to a model for SAHS protein action, and also illustrate that SAHS proteins may be practical preservatives for cells in commercial applications.

## Results

### Protein sequence analysis and expression

We selected 12 SAHS proteins from *R. varieornatus* and one from *H. exemplaris* to test for expression in the *E. coli* cytoplasm (Table S1). *R. varieornatus* is the most desiccation-resistant tardigrade and is well-characterized; a draft genome sequence is available, and crystal structures of two SAHS proteins from this organism have been solved[16,17]. We used the TargetP program[18] that computationally predicts protein subcellular localization based on the N-terminal signal peptide, which predicted that only 8 out of 12 SAHS proteins would be secreted (Table S2). However, the surprising implication that some of the SAHS proteins might not be secreted also caused us to investigate genomic regions encoding these genes.

We found that the "short" SAHS protein sequences in Uniprot that lack amino acids corresponding to the first ~40 amino acids of mature SAHS1 and SAHS4 may be incorrectly annotated; an alignment shown in Fig. 5 of Fukuda et al.[16] shows three such proteins of the 13 SAHS proteins encoded by *R. varieornatus*. For example, the DNA encoding RvSAHS9 contains a splice acceptor immediately upstream of the putative start codon, further preceded by a splice donor and an in-frame coding sequence that encodes an amino acid sequence that aligns well with the N-terminus of other SAHS proteins; RvSAHS10 and RvSAHS12 have a similar organization (Fig. 1b; Supplementary Discussion). In each case, the new putative intron has a consensus splice donor, but the polypyrimidine tract of the splice acceptor has rather few pyrimidines, which may have misled the automated systems that analyzed the genomic sequences. When spliced out, the new N-terminal amino acid sequences align well with the other SAHS mature protein N-termini. We have further identified other candidate splice sites in these genes (Supplementary Discussion).

We recombinantly expressed the Uniprot-predicted mature proteins RvSAHS1–4 and RvSAHS6-12 as well as HeSAHS4 in the *E. Coli* cytoplasm. The target proteins were expressed with an N-terminal SUMO tag, whose cleavage with a highly specific protease leaves the protein "scarless" without

any extraneous amino acids remaining (Fig. S1a)[19]. Since the N-terminal signal sequences are expected to be cleaved during secretion, we expressed proteins without these sequences based on the cleavage sites predicted by TargetP (Table S3). We successfully expressed and purified 4 proteins (RvSAHS1, 4, 6, and HeSAHS4) with high soluble expression and purity and conducted subsequent studies using them (Figs. S1, S2).

### SAHS proteins stabilize liposomes from dehydration-induced damage

As a first step of assessing the protective potential of SAHS proteins, we tested if they can stabilize lipid membranes upon dehydration-induced damages, using 1-palmitoyl-2-oleoyl-glycero-3-phosphocholine (POPC) liposomes as a model membrane. Prior to drying, the diameter of the liposomes as measured by dynamic light scattering (DLS) was in the range of 50-100 nm (Fig. 2a). Upon drying and rehydration at 1.4 mg/mL lipid concentration, less than 10% of the particles remained in this range, while larger particles appeared around 360 and 4000 nm (Fig. 2b). This marked increase in particle diameter indicated that dehydration damaged liposome membranes and promoted their fusion and/or aggregation[20].

Liposomes dried in the presence of SAHS proteins or other excipients at 10, 1, and 0.1 mg/mL concentrations were partially protected, with higher levels of excipient providing more protection. For all four SAHS proteins tested, particle sizes after a dehydration-rehydration cycle were notably smaller compared to those dried without an excipient (Figs. 2c–f and S3). While the diameters of the major peaks were slightly increased from ~60 nm to ~100 nm, none of the bigger particles having diameters larger than 200 nm were observed, which indicated that the addition of SAHS proteins prevented the fusion and aggregation of dried liposomes. Bovine serum albumin (BSA), a commonly used excipient for protein preservation, was less effective in protecting liposomes than SAHS proteins, as indicated by the signal in ~500 nm range (Fig. 2g). Liposomes dried in 10 mg/mL trehalose showed no particles over 150 nm, suggesting that trehalose also prevented membrane fusion of dried liposomes (Fig. 2h). These observations indicated that SAHS proteins have comparable effects to trehalose in terms of their ability to stabilize membranes. SAHS proteins at lower concentrations of 1 mg/mL were also able to protect liposomes, whereas at 0.1 mg/mL, the extent of protection was diminished, and larger aggregates ~1000 nm were formed (Fig. 2c–f, dotted lines); also, the repeat experiments showed consistent results (Fig. S3).

**Fig. 3 | Tardigrade SAHS proteins enhance bacterial desiccation survival. a** *E. Coli* cells were dried for 48 h with and without the addition of 0.5 mg/mL concentration of SAHS proteins, BSA and trehalose. Survival percentage was calculated by comparing the number of viable colonies after plating the rehydrated cells to the initial number of cells added. **b** *R. tropici* cells were dried for 48 h with and without the addition of 0.5 mg/mL concentration of SAHS proteins, BSA, and trehalose. Survival percentage was calculated in the same way as the *E. coli* experiment. Individual data points represent independent replicates, and lines represent the mean survival. Student's *t*-test was used to determine the statistical significance between the negative control (no excipient) and each group, which is indicated as asterisks. * $p < 0.05$; ** $p < 0.01$; *** $p < 0.001$.

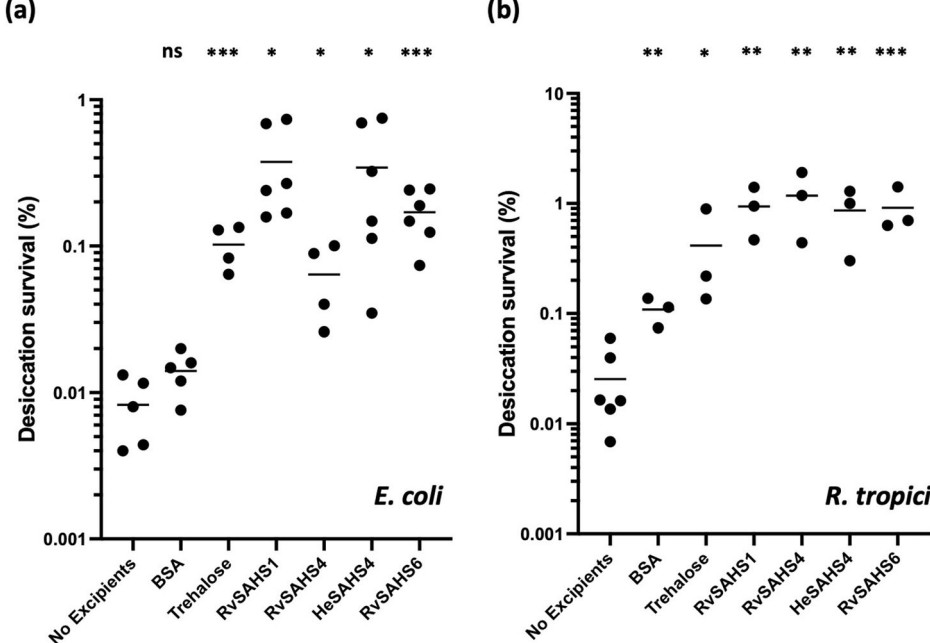

## SAHS proteins enhance survival of desiccated bacterial cells

We then tested whether extracellularly added SAHS proteins can enhance the desiccation tolerance of bacterial cells (Figs. 3 and S4). *E. Coli* cells were dried for 48 h at $10^8$ cells/mL with 0.5 mg/mL of SAHS proteins. As measured by colony-forming units, *E. coli* showed enhanced survival compared to those dried without added excipients or with BSA, lactate dehydrogenase, or citrate synthase proteins (Fig. S4a), indicating that this was not a general effect caused by adding high concentration of protein. RvSAHS1 provided the most reproducible protection of *E. coli*, with all replicates in each experiment showing >10-fold enhancement of survival. At 0.1 mg/mL RvSAHS1, the protective effect decreased but was still statistically significant (Fig. S4b). Trehalose, a well-known cryoprotectant that is expressed intracellularly in many organisms during desiccation, also protected *E. coli* against drying when added extracellularly, but not as strongly as RvSAHS1. Intracellular expression of SAHS proteins showed no clear-cut enhancement of *E. coli* survival upon desiccation (Fig. S4c, d).

We also tested if the SAHS proteins could enhance the desiccation survival of the plant-symbiotic bacterium *Rhizobium tropici*, which is a model organism for nitrogen-fixing biofertilizers. When *R. tropici* cells were dried at a concentration of $7.2 \times 10^9$ cells/mL with and without 0.5 mg/mL excipients, we observed ~40-fold enhancement of cell survival upon addition of SAHS proteins (Fig. 3b). Cells dried with trehalose at 0.5 mg/mL concentration showed 16-fold increased survival, while 0.5 mg/mL BSA caused a 4.2-fold increased survival.

## SAHS proteins and BSA protect enzymes from dehydration-induced inactivation

Tardigrade proteins such as CAHS have been shown to protect desiccated enzyme activities[21]. We tested if the SAHS proteins can preserve an enzyme, specifically lactate dehydrogenase (LDH), from desiccation-induced inactivation in vitro. When dried, LDH (0.01 mg/mL) alone lost >90% of its activity compared to the non-dried control. The four SAHS proteins helped preserve LDH activity as a function of increasing concentration from 0.001 to 5 mg/mL, but BSA's protective effect was comparable at each concentration (Figure S5). This observation contrasts with the specific effect of SAHS proteins vs. BSA on the protection of membranous structures such as liposomes and cells.

## Structural changes of SAHS proteins under desolvated conditions

To investigate the behaviors of SAHS proteins upon water loss, we examined their secondary structures using circular dichroism (CD). CD spectra of all four SAHS proteins showed minima at ~215 nm, indicating that they adopt β-structures under aqueous conditions (Fig. 4a). This result was consistent with previous literature reporting β-sheet structures in RvSAHS1 and SAHS4[14,16,17]. The CD spectra for SAHS1 and SAHS4 at 0% in aqueous solution correlate well with the solved structures of these proteins, which have a single short alpha helix and are dominated by beta strands.

To investigate if these proteins can change their structures upon environmental stress, we used increasing concentrations of trifluoroethanol (TFE) and glycerol to mimic water-deficit conditions. All four proteins remained mostly β-stranded up to 25% TFE added but displayed a notable increase in helical structures at 50% TFE. At 50–75% TFE, all proteins showed spectra with two minima around 208 and 222 nm, which are characteristic of α-helices. This result is consistent with the earlier report by Yamaguchi et al., in which RvSAHS1 adopted a more helical structure with the addition of TFE[14]. Calculating relative secondary structure content confirmed that the helical content in these proteins increased from 10% up to 40% with the addition of TFE (Fig. 4b). In contrast, increasing concentrations of glycerol had no such effect, even though glycerol is a 'crowding' agent (Fig. S6).

A filamentous, mesh-like structure was observed from dried RvSAHS1 proteins upon transmission electron microscopy (TEM) imaging (Fig. S7). RvSAHS1 proteins dried at a concentration of 1 mg/mL revealed an entangled network of fibrous structures, while those dried at a higher concentration of 10 mg/mL showed a denser network that covered most of the grid. Such observations suggested that RvSAHS1 may undergo structural changes upon loss of water to allow for higher-order structure formation.

To further investigate the pathway for structural change of SAHS proteins, we conducted a molecular dynamics (MD) simulation of the RvSAHS1 protein. To account for the possible inter-protein interactions, we first generated a water bath with three copies of a RvSAHS1 monomer, and the simulation was conducted using explicit solvent, openMM, and the Amber force field. The simulation was performed for a microsecond at 550 K to induce the breaking of hydrogen bonds in a time frame amenable to simulation. We found that the root mean squared deviation (RMSD) of alpha carbons in the ensemble relative to the starting structures increased within the first 100 ns and then plateaued for the remainder of the 1 μs (Figs. 5a, S8). As the simulation progressed, RvSAHS1 proteins showed the structural unfolding of beta-sheet structures and formation of alpha-helices: the beta-sheet component decreased from ~50% to ~3%, and the alpha-helical component increased from ~7% to ~20% as the simulation progressed from 0 to 100 ns (Fig. 5b).

Figure 5c shows the C-terminal segments of each monomer that are conserved among SAHS proteins, which contribute significantly to the initial beta-barrel structure (identified as the C3 motif in ref. 14). As the simulation progressed, this region highlighted how, as helices form, the motifs between subunits may become more closely aligned during the 0–100 ns simulation period. Figure 5d indicates how the beta-sheet region within the C3 motif, which is maintained through interactions between positively and negatively charged amino acid residues, gets separated to form a helix as the simulation proceeds. These sequences, highlighted in blue and red, were found to be highly conserved among the SAHS family (Fig. S9) based on our evolutionary conservation analysis using ConSurf[22], indicating that such structural shifts may be a shared trait among SAHS proteins.

## Structural bioinformatic analysis of SAHS proteins and related binding proteins

To understand how SAHS proteins might undergo a dramatic transition from a beta-sheet conformation to a predominantly alpha-helical state and protect vesicles and cells during drying, we examined the solved structures of RvSAHS1 and RvSAHS4, compared them with solved structures of fatty acid binding proteins from higher organisms, and analyzed patterns of sequence conservation.

The structures of RvSAHS1 and RvSAHS4 (PDB IDs 5xna and 5z4g), fatty acid binding protein from human muscle (1hmr)[23], the intracellular lipid binding protein FABP1 (6do6)[24] were examined, along with alignments of related sequences.

RvSAHS1 and RvSAHS4 are highly structured and similar to FABPs and related proteins[16,17]. Thus, the description of these proteins as "intrinsically disordered proteins" is not accurate, at least in the hydrated state. All of these proteins are unusual in that they have a large internal cavity that may be filled with ligands, solvent molecules, or both. The sides of the cavity are defined by beta sheets that, for the most part, do not pack against a typical hydrophobic core. In the RvSAHS1 and RvSAHS4 structures solved by Fukuda and colleagues, this volume is occupied by solvent molecules.

In RvSAHS1, there is a small hydrophobic core, while the cavity is bordered by beta-hairpin loops from Phe50-His72, His83-Val96, Thr106-Ile113, Ala126-Asp139, and Lys150-Tyr163. (The Phe50-His72 loop also contains two short alpha helices.) In the absence of a ligand in the cavity, both faces of these loops are simply beta-sheets that face water on both sides. Thus, these structures would not be as stabilized as in a typical protein. We hypothesize that during a desiccation process, water would equilibrate out of the cavity, and as the environment becomes more dehydrated, other more stable proteins would pack against an SAHS protein, essentially crushing it and disrupting the beta-sheet structure that surrounds the cavity. In this way, the SAHS proteins could act as deformable packing material to prevent other biological structures from deforming or rupturing during desiccation.

Among the 13 SAHS proteins of *R. varieornatus*, the amino acids pointing into the small hydrophobic center are generally conserved, while

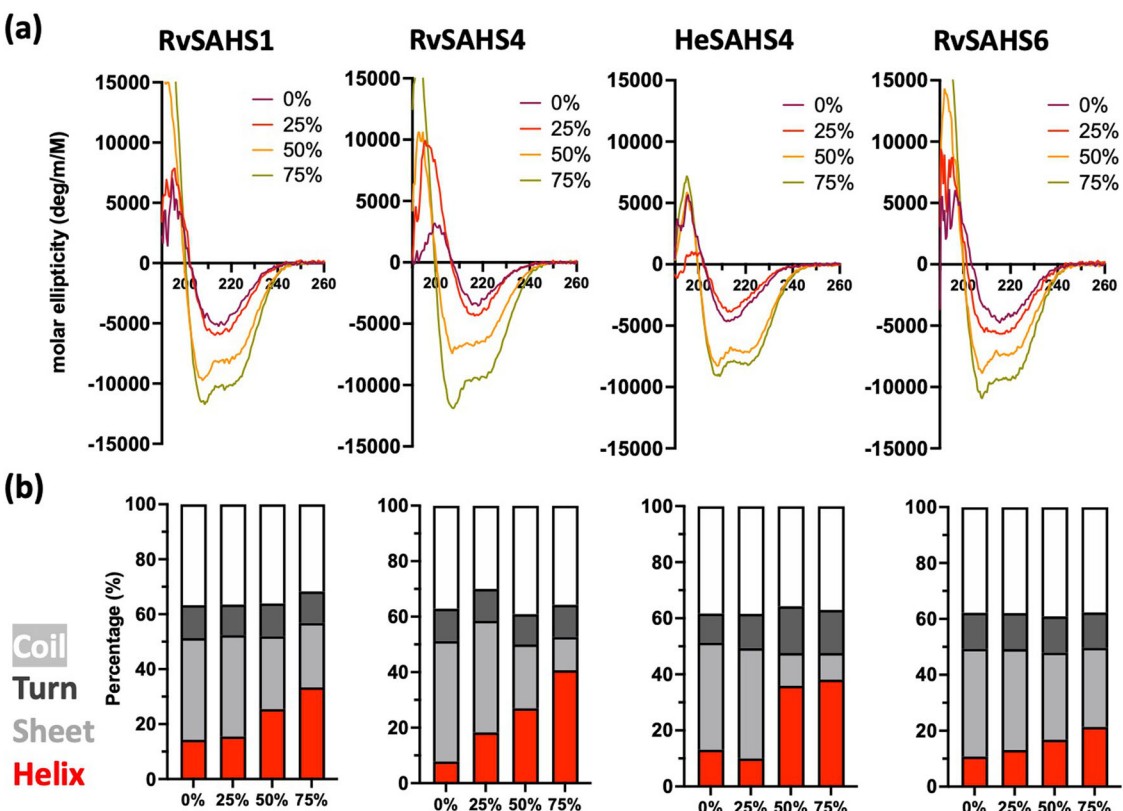

**Fig. 4 | Tardigrade SAHS proteins undergo structural changes upon dehydration-mimicking conditions. a** CD spectra of SAHS proteins upon the addition of increasing amounts of TFE from 0% to 75%. **b** Secondary structure compositions of SAHS proteins under different TFE levels, calculated from the CD spectra.

amino acids pointing into the cavity are not. The "hydrophobic core" is located on one side of the much larger interior and includes the amino acids Trp 32, Trp37, Met38, Trp41, Tyr64, Ile74, Trp75, Tyr81, Phe98, Leu100, Leu124, Tyr141, Val143, Leu148, and Ala159. Of these, Trp 32, Trp37, Met38, Trp41, Phe98, Leu100, Leu124, Tyr/Phe141, Val143, and Leu148 are rather well conserved, while the others are not. We note that the highly conserved tryptophans at positions 32, 37, and 41 are present in our newly proposed sequences for RvSAHS 9, 10, and 12, supporting the idea that these sequences were previously misannotated.

The amino acids pointing inward towards the cavity include Phe50, Phe53, Ile54, Leu57, Leu59, His83, Ile85, Val87, Lys90, Tyr92, Asp94, Thr106, His108, Leu113, Tyr115, Tyr117, Ala126, Val128, Val130, Ile137, Asp139, Lys150, Tyr152, Arg151, and Tyr165. Most of these are within 4 Angstroms of ethylene glycol or the putative fatty acid in the RvSAHS1 structure[16]. Of these, Phe53, Val/Ile87, Tyr117, Arg/Lys161 and Tyr163 are largely conserved, and the hydrophobic residues Phe50, Ile54, Ile85, Ile118, Val123, Val125, and Ile137 are generally interchangeable with other hydrophobic residues. Arg/Lys161 is notable because it is conserved in fatty acid binding proteins and interacts with the carboxyl group in the fatty acid in the same way that this amino acid interacts with a carboxyl group on a putative fatty acid in the SAHS1 structure.

Another feature of the SAHS proteins and fatty acid-binding proteins is the presence of an entry/exit point. In RvSAHS1, four turns define this area —three beta-hairpins with tips at Lys90, Asn109, and Ser132, and a one-turn alpha helix with Gln63 pointing towards the hole. The amino acid sequences around this region are extremely variable among the *R. varieornatus* SAHS proteins, with Gln63 and Ser132 near the edge of a gap in the alignment. Moreover, since the SAHS proteins are secreted eukaryotic proteins, they can undergo N-linked glycosylation at Asn-X-Ser/Thr sites (which are not captured in proteins made in *E. coli*). The distribution of putative N-linked glycosylation sites in RvSAHS1-13 is strongly biased toward the amino acids surrounding the entry/exit site: Asn109 is likely N-glycosylated in

RvSAHS1, 2, 8, 10, and 11 (and Asn107 in RvSAHS5); in RvSAHS3, 8 and 9 sites near Gln63 are likely glycosylated; and in RvSAHS5, 9 and 11 sites near Ser132 are glycosylated. RvSAHS4, 6, 12, and 13 do not have N-linked glycosylation sites. Of the 14 putative N-linked glycosylation sites in RvSAHS1-13, twelve are near the opening into the cavity. The presence/ absence and positioning of these sites may have a significant effect on the movement of water and other small molecules in and out of the cavity.

Fukuda et al. discussed the possibility that RvSAHS1 may bind a ligand with a carboxyl group (e.g. acetate) but noted that not all of the SAHS proteins have these carboxyl-binding residues[17]. Our analysis of alignments of the *R. varieornatus* SAHS proteins indicates a great deal of variation in the cavity-forming residues, implying that these proteins do not bind a common ligand (such as a fatty acid) if they bind one at all.

Based on this analysis and data presented here, we propose a model by which the thirteen SAHS proteins encoded by *R. varieornatus* undergo a transition upon dehydration in which water molecules equilibrate out of the cavity and crowding and pressure of other proteins, then disrupt the structure of the SAHS proteins, which then might reconfigure into a completely different conformation. The thirteen different SAHS proteins would have different characteristics for stability and water retention, such that they would collapse under different conditions of dehydration and pressure, allowing for a gradual process rather than a sudden phase transition. This idea is explored further in the Discussion.

## Discussion

In this study, we tested the proteins belonging to the SAHS family of tardigrade proteins for their potential to protect diverse biological structures. In addition, we observed that the SAHS proteins undergo structural shift upon exposure to a desiccation-mimicking compound and used a diverse set of experimental and computational approaches to suggest possible models of how the structural transition of SAHS proteins might occur to protect biological structures. Specifically, we found that the SAHS proteins protect

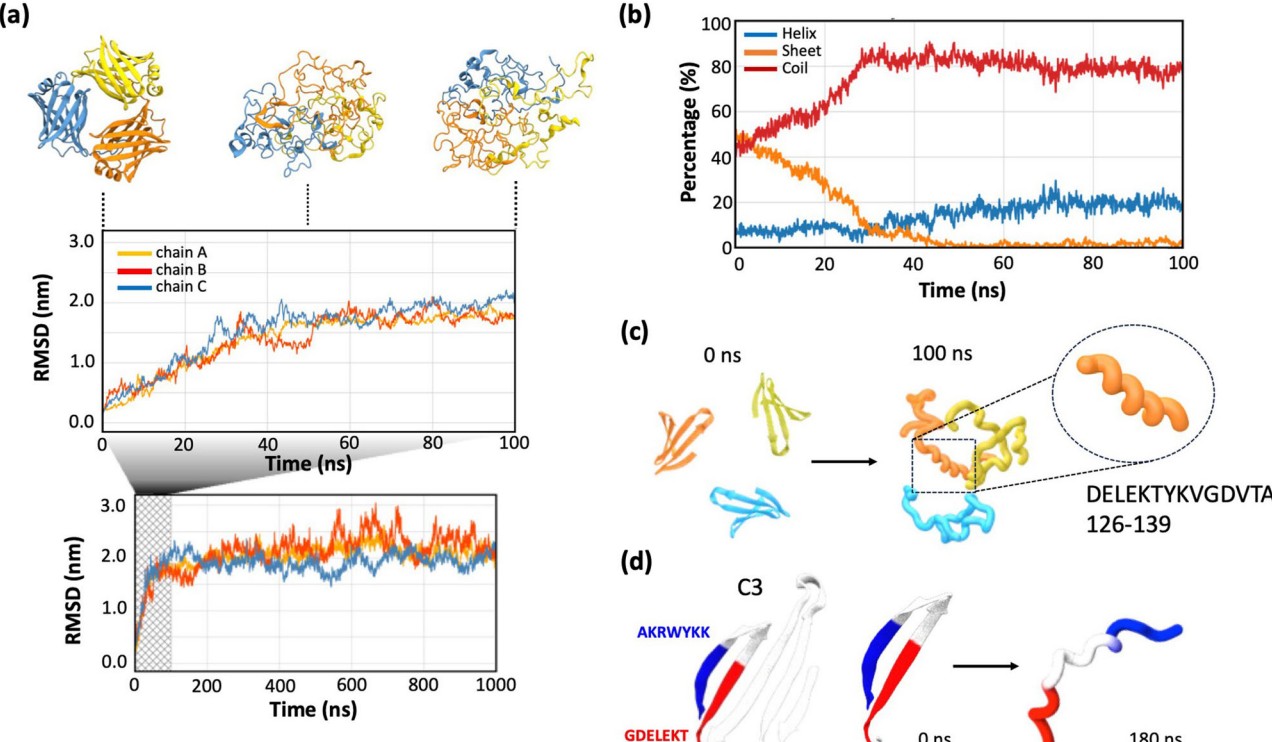

**Fig. 5 | MD simulation of SAHS protein structural change upon desolvation.** **a** RMSD over time during the first 100 ns MD simulation of RvSAHS1 homotrimer. Results during the first 100 ns are magnified. Different color indicates each monomer chain in a trimer. Representative images of the RvSAHS1 protein structures reconstituted from the MD simulation at 0, 50, and 100 ns time points are provided at the top. **b** Changes in secondary structure components of RvSAHS1 during the MD simulation. Red: coil; blue: helix; orange: sheet. **c** Representative image showing how conserved beta-sheet motifs of RvSAHS1 become closely aligned upon structural shift into helices. Only the conserved C3 segments of RvSAHS1 of each monomer are highlighted to show the changes in their alignments as the simulation progresses. **d** Detailed image of C3 motif sheet-to-helix structural change. Highlighted in blue and red are two highly conserved regions found within the beta-sheet region that directly interact through ionic bonds. Structures of these regions at 0 and 180 ns are indicated.

membrane-containing structures, specifically liposomes and living bacteria, against desiccation-induced disruption but are no better than BSA at protecting the enzyme lactate dehydrogenase against desiccation. Based on CD analysis, the SAHS proteins appear to undergo a shift from a primarily beta-sheet structure to an alpha-helical structure in increasing TFE concentrations (thought to be desiccation-mimicking conditions), and a 1 μs molecular dynamic simulation showed that short alpha-helices can transiently form upon denaturation of proteins. Lastly, we performed a comparative structural analysis based on the available putative sequences of SAHS proteins. In doing so, we found that sequences in the Uniprot database corresponding to short (N-terminally truncated) SAHS proteins may have been incorrectly annotated and that all SAHS proteins are likely secreted and share a common structure. As noted previously, SAHS proteins are distantly related to fatty acid-binding proteins and consist of a large cavity and a small hydrophobic core off to one side. Our structural analysis indicated that amino acid side chains pointing into the cavity are variable, and putative N-linked glycosylation sites are also variable in position but tend to cluster around the opening from which solvent molecules might enter and exit.

SAHS proteins from the tardigrades *R. varieornatus* and *H. exemplaris* are thought to contribute to the desiccation resistance of these organisms. *R. varieornatus* and *H. exemplaris* encode multiple different SAHS proteins, respectively, and at least some of them are constitutively expressed at high levels[13]. It is not clear why so many members of the SAHS family are encoded in these tardigrades or whether their diversity is important in desiccation protection. We tested several different SAHS proteins for their ability to be expressed in the *E. coli* cytoplasm without signal sequences. From *R. varieornatus*, SAHS1, 2, 4, 6, 7, and 8 were expressed, while the Uniprot-derived sequences of SAHS3, 9-12 were not. SAHS1, SAHS4, and SAHS6 from *R. varieornatus* and SAHS4 from *H. exemplaris* were expressed at the highest levels (Fig. S2) and were characterized further.

The four tested SAHS proteins reduced the extent of liposome fusion and bacterial loss of viability during desiccation. When a commercial POPC-based liposome preparation was dried in the absence of excipients and then rehydrated, the average diameter of the liposomes increased from about 70 nm to about 350 nm, indicating that many liposomes fuse during this process to make much larger aggregates. In the presence of 10 mg/mL of any of the SAHS proteins, the extent of liposome fusion was much reduced, and even at lower concentrations, there was a protective effect. The excipient disaccharide trehalose showed a comparable protective effect, while BSA was less protective (Figs. 2, S3). When either *E. coli* or *R. tropici* was desiccated and rehydrated, the SAHS proteins had a strong survival-promoting effect, generally improving viability by about 50-fold and performing at least 10-fold better than BSA and several other proteins and about 3-fold better than trehalose (Figs. 3, S4). In addition, heterologous intracellular expression of SAHS proteins did not significantly improve *E. coli* desiccation survival (Fig. S4) *R. tropici* is a nitrogen-fixing plant symbiont that represents a working model for the preservation of nitrogen-fixing bacteria such as rhizobia and could itself be deployed as biofertilizer.

In contrast, the SAHS proteins did not specifically protect the enzyme lactate dehydrogenase against loss of activity upon desiccation and rehydration—the SAHS proteins had a slight effect, comparable to the protective effect of BSA (Fig. S5). The ability to stabilize proteins/enzymes and prevent their aggregation has been associated with intracellular stress protection[25,26]. For instance, heterologous expressions of tardigrade small heat shock proteins (sHSPs) improved bacterial desiccation survival, and their protective mechanism has been attributed to their ability to reduce protein aggregation during desiccation[27]. Similarly, CAHS proteins, whose heterologous expressions in bacteria and yeast enhanced their desiccation survival, were superior to BSA in protecting enzymes from desiccation-induced inactivation in vitro[13,21]. The SAHS proteins tested in this study are

presumed to function extracellularly, and it might be that their main role is not to protect proteins but to prevent damage to membranes.

The solved structures of two SAHS proteins and the pattern of amino acid conservation in the SAHS proteins (Fig. 6) indicate that the SAHS proteins all have a large cavity that may simply be full of water. (An alternative hypothesis, suggested by our AlphaFold-predicted structure of multiple SAHS proteins, would be that the N-terminus of one SAHS protein fits into the cavity of another (Fig. S10); no independent evidence supports this hypothesis.) The SAHS proteins are similar in sequence to fatty acid binding proteins (FABPs), and the SAHS and FABP structures are superimposable. However, the amino acid side chains that point inward in the SAHS protein cavity are not strongly conserved, suggesting that these proteins are not binding to a particular ligand but may simply be filled with water or with variable solutes. Moreover, in FABPs the cavity is significantly larger than can be occupied by a fatty acid, and since fatty acids are moving in and out of the cavity during intracellular transport, they are presumably exchanged with water in this process.

SAHS proteins, by analogy to CAHS proteins[14], may undergo a shift to an alternative conformation in desiccation conditions. A high-temperature MD simulation of three SAHS proteins in a water box revealed that, as the beta-barrel structure of the protein became disordered, short alpha-helical regions would form and disappear (Fig. 5, S8). These structures might nucleate the formation of an alternative, protective conformation. However, the use of molecular dynamics to simulate desiccation is challenging because, during actual desiccation, the concentration of other proteins would increase and may play a dominating role in a conformational change of SAHS proteins, for example, through Brownian collisions. Moreover, desiccation occurs on long timescales that cannot be simulated with current molecular dynamics technology.

One model for the protective mechanism of SAHS proteins is that the SAHS protein cavity is filled with water and possibly other solutes in the hydrated state, but during dehydration, the water is extruded. Thus, the SAHS protein would collapse as a result of the Brownian motion of macromolecules and pressure on the outside of the protein without a corresponding pressure from the inside. This would lead to the denaturation of the protein and transition to an alternative, protective conformation. The different SAHS proteins may have different kinetics and equilibria for this transition, such that when a tardigrade is dehydrated, there is a gradual conversion of the different SAHS proteins to a protective state. The presence or absence and positioning of N-linked oligosaccharides in the different SAHS proteins near the entry/exit point may give them distinct dehydration-transition characteristics. After the conversion from a globular conformation, the SAHS proteins may form higher-order, gel-like structures that stabilize membrane-bound elements of cells, which may constitute rampart-like "special extracellular structure (SES)" observed outside of the secretory cell membranes of dried tardigrade[28].

SAHS proteins may have a practical application in protecting nontardigrade cells, such as agricultural microbes, against desiccation. We found that two bacteria, *E. coli* and *R. tropici*, could be stabilized against desiccation by the presence of a single SAHS protein. *R. tropici* is a "plant-growth promoting rhizobacterium," and PGPR use is often limited by their low desiccation tolerance[29,30], highlighting the potential applicability of SAHS proteins to improve the durability of microbial biofertilizers. Also, desiccation would enable easier and lower-cost transportation and safer handling of these microbes in the field. In summary, this work provides the evaluation of tardigrade SAHS proteins in terms of their protective abilities, which supports their potential applications as stabilizing molecules.

## Methods
### Sequence analysis
SAHS protein sequences used for expression in *E. coli* (Figure S2) were obtained from Uniprot database (https://www.uniprot.org). Protein subcellular localization and signal peptides were predicted using TargetP 2.0 server available at https://services.healthtech.dtu.dk[18]. Genomic sequences encoding *R. varieornatus* SAHS proteins were from the *Ramazzottius varieornatus* contig Scaffold001, strain YOKOZUNA-1, whole genome shotgun sequence, GenBank accession number BDGG01000001, which encodes SAHS1-12, and *Ramazzottius varieornatus* contig Scaffold002, GenBank accession number BDGG01000002, which encodes SAHS 13.

### Protein expression and purification
Expression vectors encoding His$_6$/SUMO-tagged SAHS proteins were obtained from Twist Bioscience and transformed into LEMO21(DE3) competent cells (New England Biolabs) according to the provider's protocol. Transformed cells were grown in TBM-5052 autoinduction media containing 100 µg/mL Kanamycin, 25 µg/mL Chloramphenicol, and 2 mM L-rhamnose at 37 °C overnight. The cells were harvested by centrifugation at 4000×*g* for 10 min, resuspended in lysis buffer (20 mM NaH$_2$PO$_4$, 500 mM NaCl, 20 mM Imidazole, pH 7.5) supplemented with 0.25 mg/mL lysozyme (Millipore Sigma), turbonuclease (Accelagen) and 1% n-Nonyl-Beta-D-Glucopyranoside (Cube Biotech) and shaken vigorously for 1 h. The total lysate was then centrifuged at 4000×*g* for 30 min, and the supernatant was obtained. This clarified protein lysate was then shaken with Ni-charged IMAC Magbeads (Genscript) for 1 h to bind tagged proteins. Beads were then washed 3 times with wash buffer (20 mM NaH$_2$PO$_4$, 500 mM NaCl, 20 mM Imidazole, pH 7.5), washed once more with PBS buffer, and incubated with PBS buffer supplemented with Cth SUMO protease[19] at 0.02 mg/mL shaking overnight to achieve the cleavage of SAHS proteins from N-terminal SUMO tag. Cleaved proteins were recovered from the supernatant, filtered through 0.2 µm filters, concentrated using Amicon ultra 0.5 centrifugal columns (Millipore, 10 kDa MWCO), and stored at −20 °C. Purified proteins were inspected using SDS–PAGE and SimplyBlue staining (Invitrogen).

### Circular dichroism
Proteins were buffer exchanged by overnight dialysis with 20 mM NaH$_2$PO$_4$, pH 7.5, and diluted to a final concentration of 0.25 mg/mL. Protein secondary structure was examined using far-UV circular dichroism (CD) using a Jasco J-815 CD spectropolarimeter equipped with a Peltier temperature controller and single cuvette holder. The CD spectra were obtained by averaging three wavelength scans from 200 to 260 nm in 0.5 nm steps in a cuvette with a path length of 1 mm. To examine protein secondary structures in water loss-mimicking conditions and molecular crowding conditions, trifluoroethanol (TFE) or glycerol was added to the sample at final concentrations ranging from 25% to 75% and incubated for two hours before determining the CD spectra.

### Lactate dehydrogenase inactivation assay
L-lactate dehydrogenase (LDH) enzyme (Roche) was diluted to 0.01 mg/mL concentration in 100 µL of 25 mM Tris/HCl buffer containing various concentrations of SAHS proteins or BSA (Sigma-Aldrich). BSA were dialyzed against the same buffer prior to use. 50 µL of each sample was stored at 4 °C while the other 50 µL was dried using a Savant SPD131DDA SpeedVac (Thermo Scientific) for 2 h without heating. Dried samples were rehydrated, and activity was measured at 50× dilution in a buffer containing 100 mM sodium phosphate (pH 6), 100 µM NADH, and 2 mM pyruvate. Absorbance at 340 nm was measured using a Synergy H1 plate reader (BioTek). Relative activity after dehydration was calculated by comparing the initial linear reaction rate of the dried sample to that of the non-dried control sample.

### Liposome drying assay
POPC liposomes (100 nm, Tribioscience) dialyzed against PBS buffer were diluted to 1.4 mg/mL in 50 µL PBS buffer containing various concentrations of proteins and excipients. Proteins were dialyzed against the PBS buffer prior to use. Samples were then dried in a 250 µL microcentrifuge tube with an opened lid placed inside a 750-mL sealed chamber filled with about 42 grams of Drierite desiccant for 48 h at room temperature. Samples were rehydrated and diluted 10-fold, and the liposome sizes were measured using a Wyatt DynaPro Plate Reader III dynamic light scattering (DLS) device.

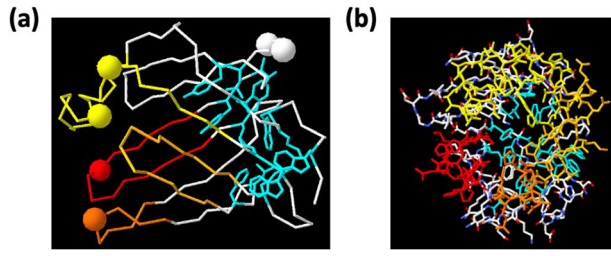

**Fig. 6 | Glycosylation sites in the SAHS proteins of *R. varieornatus* cluster near the cavity opening. a** Structure of SAHS1 from Fukuda et al.[16], showing an alpha-carbon trace of all amino acids, with the beta-hairpins surrounding the cavity colored in yellow, yellow-orange, orange and red. Amino acids in cyan with side chains shown define the hydrophobic core. Large spheres show the alpha carbons of asparagines in an Asn-X-Ser/Thr sequence in at least one SAHS protein. The putative opening of the cavity is on the left. **b** Structure of SAHS1, rotated 90 °C relative to panel A with the left side coming forward. All the side chains are shown. This view illustrates the opening into the cavity of the protein, such that the cyan-colored hydrophobic core side chains at the back of the protein can be seen through this hole.

## Bacterial cell drying assay

*E. coli* vectors expressing SAHS proteins without SUMO tags and with an N-terminal methionine were custom-ordered and obtained from Twist Bioscience and transformed into LEMO21(DE3) competent cells (NEB). To test for the effects of intracellular SAHS expression, cells transformed with each SAHS protein as well as mEGFP expression vector were grown in LB medium containing 100 µg/mL Kanamycin and 25 µg/mL Chloramphenicol at 37 °C until OD600 of 0.6, added with 0.1 mM IPTG and 2 mM L-rhamnose and further grown at room temperature overnight. Cells were harvested and resuspended in PBS buffer and diluted to contain $5 \times 10^6$ cells in 50 µL PBS buffer. Cells were then dried in a microcentrifuge tube with an opened lid placed inside a sealed chamber filled with Drierite desiccant for 48 h at room temperature. Subsequently, cells were rehydrated and immediately plated onto LB-agar Kan/Cm selection plates with a serial dilution to count the colony-forming units the next day. To test for protection by extracellular SAHS proteins, LEMO21(DE3) cells transformed with mEGFP expression vector were grown in LB medium containing 100 µg/mL Kanamycin, 25 µg/mL Chloramphenicol at 37 °C overnight, without any rhamnose and inducer added. Subsequently, cells were harvested and resuspended in PBS buffer and diluted to contain $5 \times 10^6$ cells in 50 µL PBS buffer supplemented with varying concentrations of SAHS proteins or other control excipients, dried, and rehydrated following the above protocol. BSA and excipients were dialyzed against the PBS buffer prior to use.

For *R. tropici* drying assay, *R. tropici* CIAT 899 cells were grown in PY media overnight in round-bottom culture tubes. The culture tubes were placed in an incubator set to 28 °C and shaken at 200 rpm, then transferred to larger volume flasks the following day. Following incubation, the bacterial culture was centrifuged at 4000×*g* for 15 min. The resulting pellets were then resuspended in phosphate-buffered saline (PBS) and diluted to reach an OD600 of 0.4. The cultures were centrifuged again, and the supernatant was replaced with PBS buffer containing excipients. The solutions were pipetted into 250 µL tubes in 100 µL aliquots to contain a total of $7.2 \times 10^8$ cells and placed inside a sealed container with Drierite desiccant (10–20 mesh, Thermo Scientific) to allow the samples to dry for 48 h. Once the samples were dried, three samples of each condition were taken out at relevant time points and resuspended in PBS. Subsequently, the samples were diluted and spread onto PY agar plates for colony counting.

## Transmission electron microscopy (TEM)

SAHS proteins in PBS buffer were imaged by transmission electron microscopy using a Tecnai G2 Spirit BioTWIN TEM equipped with an AMT 2k CCD camera. TEM samples were prepared by depositing proteins onto carbon/formvar coated copper grids, staining with 2% uranyl acetate solution.

## Computational modeling and conservation analysis of RvSAHS1

An initial structure for RvSAHS1 was generated using alphaFold using the sequence acquired from UniProt with the accession code J7MFT5. Ambertools generated amber input files with the ff19SB forcefield and explicit TIP3 solvent conditions. OpenMM was used to simulate 1000 ns at a temperature of 550 K after an equilibration at 310 K for 1 ns. Trajectories were generated with conformations captured every 10 ps. RMSD analysis of the simulation trajectory was performed using the Python package mdtraj. Visualizations of protein structures were produced using Houdini (SideFX software) and the Python packages mdtraj, biopython, and prody. Additional details of computational methods are in Supplementary Information, and structural details are included in Supplementary Data 2.

Evolutionary conservation analysis was performed on the RvSAHS1 sequence (J7MFT5) using ConSurf and the default parameters, where 60 homologs were collected from the UNIREF90 database using HMMER. Of these, 24 homologs passed the default thresholds (min/max similarity, coverage, etc.), and 25 were CD-HIT unique. The conservation calculations were conducted on the 25 unique hits, including the initial query.

## Statistics and reproducibility

For the desiccation survival experiments, the Student's *t*-test was used to determine the statistical significance between the negative control (no excipient) and cells dried with SAHS proteins or excipients. The number of independent replicates is noted in the figure legend and separately attached source data. For the enzyme activity assay, the number of replicates is noted in the figure legend and source data.

## Reporting summary

Further information on research design is available in the Nature Portfolio Reporting Summary linked to this article.

## Data availability

All data are either contained within the manuscript/supporting information or available from the authors upon request. Source data for figures are included in Supplementary Data 1. Supplementary Data 2 includes the initial structural pdb file of the MD simulation and those at time points of 50, 100, 150, 200, 250, and 300 ns. Bacterial expression plasmids for the production of recombinant SAHS proteins can be acquired from Addgene (Addgene plasmid ID 215645–215648).

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

## Acknowledgements

This work was supported by the Wyss Institute for Biologically Inspired Engineering at Harvard University as the validation project. This work was further supported by the DARPA under Cooperative Agreement Number W911NF-19-2-0017. The authors acknowledge the Harvard Medical School Electron Microscopy Facility for the use of TEM and the Center for Macro-molecular Interactions at Harvard Medical School for the use of CD and DLS. B.M. acknowledges NSF CMMI-1752172 and ONR N000141912317. Z.B. acknowledges support from the NSF Graduate Research Fellowship Program.

## Author contributions

S.L. designed the research, performed the experiments, analyzed the data, and wrote the paper. C.B.R. conducted the molecular dynamics simulation and corresponding analysis. Z.B. and B.M. performed and analyzed the *R. tropici* experiments. J.C.W. performed structural analysis, analyzed the data and wrote the paper. P.A.S. analyzed the data, contributed to experiment design and edited the paper.

## Competing interests

The authors declare no competing interests.
