## [Peer Review File · Communications Biology]

Reviewers' comments:

Reviewer #1 (Remarks to the Author):

Because water is essential for all organisms, terrestrial organisms have systems to struggle with desiccation stresses. For example, tardigrades can survive severe dried environments by anhydrobiosis or "life without water". Although their extraordinary ability is very popular, little is known about detailed molecular bases of anhydrobiosis. In 2012, Yamaguchi et al. discovered two tardigrade specific proteins which are highly expressed under desiccated conditions: secretory abundant heat soluble (SAHS) and cytoplasmic abundant heat soluble (CAHS) proteins. Later genomic and biochemical analyses have further supported that these proteins are involved in desiccation tolerance of Heterotardigrada. Because crystal structures of SAHS proteins revealed their structural similarity to fatty acid binding proteins (FABPs), they are thought to store or transfer hydrophobic biomolecules like FABPs. However, how SAHS proteins can contribute to desiccation tolerance have been largely unknown.

The manuscript "Tardigrade secretory proteins protect biological structures from desiccation" by Lim et al. reports biochemical and computational studies that support an interesting new hypothesis about SAHS proteins. They suggest that SAHS proteins undergo a structural change from β -sheet-rich to α -helix-rich structures and this transformed structure can protect biological structures such as liposomes and cells. This hypothesis is novel and the experimental procedures to support it have been conducted properly.

Although I don't think this fascinating hypothesis is absolutely correct, I believe that providing a variety of hypotheses to the research community is essential to solving the big mystery of tardigrade anhydrobiosis. Besides, this manuscript provides revised amino acid sequences of several SAHS proteins. This information is important for researchers in the related fields. Therefore, I strongly recommend publication of this manuscript in *Communication Biology*.

Questions and comments

Page 1. Abstract. Line 19.

The original report on SAHS proteins by Yamaguchi et al. uses "secretory abundant heat soluble", not "secreted-abundant heat soluble".

Page 3. Line 53.

Are positions of references 10 and 11 correct? These papers seem to describe the relationship between tardigrades and trehalose, not LEA proteins or TDPs.

Page 3. Line 56.

"secreted-" should be "secretory-".

Page 7. Line 140 (Page 8 line 164).

Figure S5 appears in the main text before Figure S4.

Page 8 Structural changes of SAHS proteins under desolved conditions.

1) Yamaguchi et al. has already performed a similar CD experiment and obtained a similar result. Please mention this earlier result.

2) SAHS proteins are classified as lipocalin proteins. It is well-known that lipocalin proteins (and many other proteins) undergo β -to- α transition by addition of TFE that provides dehydrated conditions (e.g. Shiraki et al. *J Mol Biol.* 245(2), 180–194 (1995); Konno. *Protein Sci.* 7(4), 975–982 (1998); Kumar et al. *Biochemistry* 42(46), 13708–13716 (2003)). Therefore, the transition reported in this manuscript itself is not specific to SAHS proteins.

Therefore, I have a few questions. Do other lipocalin proteins such as FABPs have a similar biological structure protecting ability? Some FABPs are known to be secreted to extracellular regions (e.g.

Villeneuve et al. *J Cell Biol.* 217(2), 649–665 (2018)). If only SAHS proteins can perform protection of liposomes and cells, why?

3) line 180. The tardigrade cellular and extracellular regions must be crowding environments especially under dried conditions. But TFE induced no or negligible β -to- α transition under such conditions. Can the β -to- α transition of SAHS proteins occur in real tardigrade bodies?

4) line 182. The authors could observe mesh-like and network structures of SAHS proteins dried on TEM grids. Do other proteins form such structures under the same condition?

5) The authors only used RvSAHS1 for MD simulation. Because they could use an AlphaFold structure as a starting model, they could have tried simulations on other SAHS proteins and other lipocalin proteins. Why did they only use RvSAHS1?

6) As for the details of MD simulation. In the method section, the authors say, "An initial structure for RvSAHS1 was generated using AlphaFold...". However, there are many versions of AlphaFold (original AlphaFold2 by Jumper et al., ColabFold, etc..) and they show different performances. For better reproducibility, please add the detail. Also, it is unclear how they put three molecules in the water bath and why they use just three SAHS molecules, not one, two, or more. Providing the starting structure coordinates or something as a supporting material can be another solution. If I'm correct, the reference for AlphaFold is missing in the manuscript.

5) line 191. Does "using explicit solvent" mean that no water molecules bind to protein surfaces in this MD calculation? If so, such a situation is far from the real environments. In fact, even in the anhydrobiotic state, organism bodies contain residual water (Potts. *Microbiol Rev.* 58, 755–805 (1994)) and some tardigrade proteins such as CAHS proteins show high affinity to water (Arakawa and Numata. *Mol Cell.* 81(3), 409–410 (2021)). Secondary structure percentages are quite different between the MD simulation and the CD experiments. Please add a reasonable explanation about this gap.

6) Page 9. line 207.
Does Figure S9 mean Figure S8 (C)?

7) Page 10. line 208.
One of two conserved region mentioned here seem to be just a ligand binding site conserved among FABPs and FABP-like proteins. Therefore, again, I wonder why only SAHS proteins have a protecting ability.

Page 12. Line 274.
The authors suggested that SAHS proteins do not bind a common ligand. What does "common ligand" mean here? To some extent, FABPs, FABP-like proteins, and other lipocalin proteins show ligand selectivity and bind different ligands. They do not have to bind a common ligand. That's why they show amino acid sequence variations.

Reviewer #2 (Remarks to the Author):

The manuscript (MS) by Lim et al. presents an experimental and computational study of secreted-abundant heat soluble (SAHS) proteins, one of the three types of proteins involved in the ability of tardigrades to survive in complete desiccation. The investigation of tardigrade proteins related to their capability to endure extreme stresses has aroused a great interest for their potential biotechnological and biomedical applications. This MS represents a novel contribution to this research by showing the role of SAHS proteins in protecting biomolecules from desiccation-induced damage and enhancing

survival of desiccated bacterial cells.

The MS is well presented and well written. The experimental part investigating in vitro the protection effects of SAHS is correctly addressed and discussed. However, the computational part addressing with MD calculations an alleged structural change of SAHS proteins under desiccation suggested by CD spectra needs more elaboration. The protective mechanism associated with that structural change is "envisioned" (authors's term) without a firm enough basis. My comments below address these impressions alongside other remarks.

1. In lines 55-56, the authors state that the disorder in TDPs is an "inference". It should be pointed out that there currently are reliable predictors of protein disorder from sequence (doi.org/10.1038/s41592-021-01117-3). Structural disorder presented in the literature is in most cases directly predicted from sequence. In fact, for the three types of TDPs required for desiccation tolerance in the tardigrade *Hypsibius dujardini* presented in ref.15, disorder is directly based on the systematic prediction with IUPred (doi.org/10.1093/nar/gkab408) for the sequences of the 58 CAHS, 5 MAHS, and 42 SAHS studied there.

2. In lines 64-65 and then in lines 221-222, the authors state that describing SAHS proteins as intrinsically disordered proteins is not accurate. This is correct but it should be qualified by pointing out that whereas CAHS and MAHS are shown in ref. 15 to be completely or largely disordered, the 42 predictions for all the SAHS in the supplementary information of ref. 15 reveal that they are nearly completely ordered. Although this study refers to *H. dujardini*, its results could reasonably be assumed also valid for *Ramazzottius varieornatus*.

3. In the paragraph introducing the structural features of SAHS proteins (lines 61-69), the authors refer to the crystal structures of RvSAHS1 and RvSAHS4. It should be also mentioned that there are predicted structures in the AlphaFold Protein Structure Database for all the SAHS proteins (12 from *R. varieornatus* and 1 from *H. dujardini*) studied in the MS. The comparison between crystal and AlphaFold structures of RvSAHS1 and RvSAHS4 demonstrates the high reliability of the predicted structures and provides a basis for the compared structural study mentioned in points 4 and 9 below.

4. Since there are 3D model structures for the 13 SAHS studied, it should be interesting to compare them with each other to detect differences that could be relevant to some of the issues addressed in the MS such as for instance, their (apparently) distinct subcellular localization or the different size of their internal cavity. While the authors address this issue by studying sequence (pages 4 and 5), it might be interesting to do it by studying structure. In this regard, they can check that in Dali (<http://ekhidna2.biocenter.helsinki.fi/dali/>) the tool "All against all" provides a correspondence analysis plot and a dendrogram in which the 13 SAHS are organized in different groups on the basis of the slight structural differences detected in their 13 AlphaFold models.

5. Given that there are crystal structures for RvSAHS1 and RvSAHS4, it should be interesting to compare the percentages of the secondary structure elements obtained from CD spectra in the 0% case in Figures 4B and S6 with that directly obtained from the crystal structures. That comparison might provide an assessment for the quantitative considerations in lines 195-199. Although one could argue that the environmental conditions of proteins for X-ray diffraction and CD work are different, this also happens for CD measurements and MD simulations that are used in the MS to address the structural change of RvSAHS1.

6. A filamentous, fibrous network structure is obtained in TEM images for RvSAHS1 dried at two distinct concentrations (Fig. S7). This interesting result is succinctly (lines 182-187) presented but it deserves further highlighting. An analogous finding for TDPs is addressed in ref. 4 (its very title announces it) but the authors do not mention it. Also published in 2022 is other article (not referenced in the MS) that deals with assembly of TDPs into fibrous gels in response to environmental stress (doi.org/10.1002/anie.202280161).

7. The MD simulation on RvSAHS1 should be considerably clarified. Not only their methodological details are unclear, but it should be explained how a MD simulation on a system prepared with the protein properly solvated is used to study the protein in desiccation conditions. Specifically, the following points should be addressed. (a) An initial structure is generated from sequence "...using alphaFold" (lines 506-507) but not details about the calculation are presented if this is done with AlphaFold2 software (which is not properly referenced). Given that this structure is available in the AlphaFold Database (which is not properly referenced, neither), why did the authors generate it? (b) No details about the solvation box other than the TIP3 water model is used (line 508). (c) If the solvated structure is equilibrated at 310 K for 1 ns, how is the thermal connection with the simulation at 550 K done? (d) No details about the ensemble in which the simulation is performed are given. (e) No details about the methods to control T and P constant (if the NPT ensemble is used) are given. (f) No details about the truncation of nonbonded pair VdW interactions and long-range electrostatic interactions are given. (g) It is stated (line 192) that the simulation is performed at 550 K "...to induce breaking of hydrogen bonds", yet at that temperature there is no liquid water. (h) No details about the method used to identify secondary structure along the simulation (Fig. 5B) are given. (i) If H-bonds are not present due to the high temperature, one cannot speak of "alpha helices" or "beta sheets". One should instead speak of (Phi, Psi) dihedral angles reminiscent of alpha helix or beta strand conformations. But then a method to compute those angles along the simulation must be presented.

8. Irrespective of the concerns raised in the preceding point, the main effect seen in Fig. 5B is the change of beta sheet to coil: orange and red plots are nearly specular images. Compared to this, the slight increase shown by the blue plot (alpha helix) is a secondary effect. Moreover, the structure at 100 ns in Fig. 5C shows a clearly helical arrangement only in chain A (orange) while chains B and C exhibit irregular conformations without a clear helical pattern. In view of these results, the claims that "...SAHS proteins might undergo a dramatic transition from a beta sheet conformation to a predominantly alpha-helical state" (lines 212-213) or that (SAHS proteins) "...reconfigure into a completely different conformation dominated by alpha-helices" (lines 279-280) or "...shift from a primarily beta-sheet structure to an alpha-helical structure" (lines 293-294) seem slightly exaggerated. In this regard, the conjectural statements about SAHS proteins acting as deformable packing material (lines 230-236) speak much more appropriately of "disrupting the beta sheet structure" (line 234).

9. The discussion on the organization of amino acids with respect to the cavity and its entry/exit point compared with FABPs is interesting and provides information useful to unveil the possible movement of water in the changes occurring under desiccation. The results of the comparison provided by Dali mentioned in point 4 above showing slight differences in the size of the cavity in some of the 13 SAHS proteins could be probably relevant here.

10. In lines 305-314, the authors highlight *R. varieornatus* as compared with *H. exemplaris* with regard to the large number of SAHS proteins wondering why the former tardigrade encodes as large SAHS members as 13. They should remark that, as mentioned in points 1 and 2 above, ref. 15 presents 42 SAHS members in *H. dujardini*.

11. Incidentally, terms *Hypsibius dujardini* and *Hypsibius exemplaris* are frequently used in the literature as synonymous and this MS also does it. But strictly they are not: see doi.org/10.11646/zootaxa.4415.1.2.

The Discussion presents a valuable synthesis of the results on RvSAHS proteins reported in the MS. The interesting findings that two bacteria could be stabilized against desiccation by a single SAHS protein is properly emphasized and a novel mechanism for the protective action of SAHS proteins is also tentatively proposed. Except the recurrent claims that SAHS proteins adopt alpha-helical structures when the evidence presented rather suggests that beta-sheet structures are just unfolded, the model envisioned (their term) for that mechanism together with the experimental results make

this MS an interesting and original contribution. Provided that the concerns raised in this review are properly addressed, this work could be relevant for researchers of tardigrade proteins involved in the outstanding abilities of these animals to survive a wide range of environmental stresses.

Point-by-point response to reviewers' comments.

Reviewer 1

Because water is essential for all organisms, terrestrial organisms have systems to struggle with desiccation stresses. . . . [Reviewer 1 summarizes our manuscript.] Therefore, I strongly recommend publication of this manuscript in Communication Biology.

- *We appreciate the reviewer's positive response.*

Questions and comments

Page 1. Abstract. Line 19.

The original report on SAHS proteins by Yamaguchi et al. uses "secretory abundant heat soluble", not "secreted-abundant heat soluble".

- *We now use the term "secretory abundant heat soluble" throughout the manuscript (lines 20 and 62).*

Page 3. Line 53.

Are positions of references 10 and 11 correct? These papers seem to describe the relationship between tardigrades and trehalose, not LEA proteins or TDPs.

- *We have edited the text to refer to references 10 and 11 in the context of tardigrade survival in extreme conditions, and reference 12 is cited in the context of tardigrade-specific survival proteins. (lines 57-58).*

Page 3. Line 56.

"secreted-" should be "secretory-".

- *We have changed the wording. (Line 62).*

Page 7. Line 140 (Page 8 line 164).

Figure S5 appears in the main text before Figure S4.

- *We have changed the figure numbering.*

Page 8 Structural changes of SAHS proteins under desolved conditions.

1) Yamaguchi et al. has already performed a similar CD experiment and obtained a similar result. Please mention this earlier result.

- *We now mention and cite (ref. 13) Yamaguchi et al.'s earlier results in this section. (Lines 243-245)*

2) SAHS proteins are classified as lipocalin proteins. It is well-known that lipocalin proteins (and many other proteins) undergo β -to- α transition by addition of TFE that

provides dehydrated conditions (e.g. Shiraki et al. J Mol Biol. 245(2), 180–194 (1995); Konno. Protein Sci. 7(4), 975–982 (1998); Kumar et al. Biochemistry 42(46), 13708–13716 (2003)). Therefore, the transition reported in this manuscript itself is not specific to SAHS proteins.

Therefore, I have a few questions. Do other lipocalin proteins such as FABPs have a similar biological structure protecting ability? Some FABPs are known to be secreted to extracellular regions (e.g. Villeneuve et al. J Cell Biol. 217(2), 649–665 (2018)). If only SAHS proteins can perform protection of liposomes and cells, why?

-To the best of our knowledge, no study has directly tested the ability of other lipocalin proteins to stabilize biological structures under drying conditions using similar experiments to this study. Experimentally verifying protective abilities of proteins other than SAHS family members is outside the scope of this work. We also note that the reviewer-cited proteins are from organisms that are not desiccation-tolerant. The β -to- α transition may contribute to the protective capability of SAHS proteins, but may or may not be sufficient for it.

-We note that several papers reported that lipocalin proteins such as α_1 -acid glycoprotein (AGP) or tear lipocalin (TL) can undergo β -to- α transition to interact with membranes (Ruiz, Front. Physiol. 12, 686251 (2021); Gasymov et al. Biochim. Biophys. Acta, Protein Struct. Mol. Enzymol. 1386, 145-156 (1998)). In addition, membrane maintenance has been suggested as one of the ancestral functions of lipocalin family (Ganforina et al. Front. Physiol. 13, 904702 (2022)).

-We appreciate that the reviewer appears to be asking good questions but does not suggest changes in the manuscript.

3) line 180. The tardigrade cellular and extracellular regions must be crowding environments especially under dried conditions. But TFE induced no or negligible β -to- α transition under such conditions. Can the β -to- α transition of SAHS proteins occur in real tardigrade bodies?

- We think that the reviewer is referring to Figure S6, in which glycerol, rather than TFE, is used as a 'crowding' agent – TFE strongly induces a β -to- α transition (Figure 4). High concentrations of glycerol indeed have no effect on α -helical content. This negative result is included in Supplementary Information because it may constrain models of SAHS protein structural transition. It is possible that glycerol can replace water in the cavity of SAHS proteins and protects them against a structural transition, or at least does not promote it. We note that glycerol may function as a 'crowding' agent by essentially titrating water so that ionic strength is effectively increased, but this is different from crowding of proteins; i.e. glycerol addition may remove water but not change the amount of bulk solvent.

4) line 182. The authors could observe mesh-like and network structures of SAHS proteins dried on TEM grids. Do other proteins form such structures under the same condition?

- Recent publications reported that other tardigrade proteins are capable of forming filamentous structures under dehydrated conditions. For instance, tardigrade cytosolic abundant heat-soluble (CAHS) proteins formed filamentous structures both in vitro and in vivo (Yagi-Utsumi et al. Sci. Rep. 11, 21328 (2021); Tanaka et al. PLOS Biol. 20, e3001780 (2022); Malki et al.

Angew. Chem. Int. Ed. 61, e202109961 (2022).). In particular, Yagi-Utsumi et al. reported CAHS protein fibers dried on TEM grids resembling our results, and Malki et al. reported formation of porous mesh-like structures at high protein concentration (30 mg mL⁻¹), which was consistent with our results.

- We now mention these points in the legend to Figure S7, where the TEM images are shown.

5) The authors only used RvSAHS1 for MD simulation. Because they could use an AlphaFold structure as a starting model, they could have tried simulations on other SAHS proteins and other lipocalin proteins. Why did they only use RvSAHS1?

- The process we are trying to understand represents challenges for simulation that we do not know how to address, and to the best of our knowledge are not addressible with current simulation technology. We now discuss these limitations explicitly in the new section on simulation methods in the Supplementary Information (p. 22). We also felt that performing MD simulation for every other AlphaFold-based SAHS protein variant would be redundant, especially considering the amount of computational resources required for performing each 1 microsecond simulation.

6) As for the details of MD simulation. In the method section, the authors say, “An initial structure for RvSAHS1 was generated using AlphaFold...”. However, there are many versions of AlphaFold (original AlphaFold2 by Jumper et al., ColabFold, etc...) and they show different performances. For better reproducibility, please add the detail. Also, it is unclear how they put three molecules in the water bath and why they use just three SAHS molecules, not one, two, or more. Providing the starting structure coordinates or something as a supporting material can be another solution. If I'm correct, the reference for AlphaFold is missing in the manuscript.

AlphaFold2(Jumper, John, Richard Evans, Alexander Pritzel, Tim Green, Michael Figurnov, Olaf Ronneberger, Kathryn Tunyasuvunakool, et al. 2021. “Highly Accurate Protein Structure Prediction with AlphaFold.” *Nature*, July. <https://doi.org/10.1038/s41586-021-03819-2>. (<https://github.com/google-deepmind/alphafold>) was used and is now cited in the new section in Supplementary Information. The starting structure is provided as a Supplementary File, as well as structures for intermediate timepoints in the simulation.

Initially, predicted structures were generated for 1, 2, 3, 4, 8, and 16 molecules, and various structural complexes were revealed. The choice of using three for simulation was to strike a balance between capturing protein-protein interactions while not building an inefficient simulation system for long timescales. The choice of using three SAHS1 molecules in the simulation was also to have multiple proteins whose behavior could be compared. This information is now supplied in the new computational simulation methods section.

In addition, part of the rationale for generating predicted structures based on more than one copy of SAHS1 was to see if AlphaFold might predict an interaction between subunits. In fact, AlphaFold did predict that the N-terminus of one SAHS1 subunit could enter the cavity of a second subunit. This was depicted in a figure in the original manuscript but not noted because we had no independent experimental data indicating that this interaction actually occurs. Both reviewers are interested in our modeling observations, so we have now described this potential interaction in Supplementary Figure S10.

7) line 191. Does “using explicit solvent” mean that no water molecules bind to protein surfaces in this MD calculation? If so, such a situation is far from the real environments. In fact, even in the anhydrobiotic state, organism bodies contain residual water (Potts. Microbiol Rev. 58, 755–805 (1994)) and some tardigrade proteins such as CAHS proteins show high affinity to water (Arakawa and Numata. Mol Cell. 81(3), 409–410 (2021)).

Secondary structure percentages are quite different between the MD simulation and the CD experiments. Please add a reasonable explanation about this gap.

-Explicit solvent (i.e. water) was incorporated in our simulation. While desiccation removes most of the water, other molecules remain present and provide a variety of inter-molecular interactions that influence intra-molecular interactions. This approach contrasts with other studies that simulate proteins in a vacuum, a method noted by the reviewer as not physiologically relevant. Desiccation subjects proteins to "extreme conditions," including fluctuations in pressure, salt concentrations, and mechanical forces. While replicating all these conditions is beyond the scope of this study, we have attempted to mimic such extreme conditions by employing high temperatures while maintaining pressure. At these elevated temperatures, hydrogen bonds break or break and reform more quickly, and the protein's intramolecular structure is disrupted, yet water molecules continue to interact with the protein.

-The CD study was performed on samples that had been incubated for 2 hours in trifluoroethanol (which we now clarify in the Methods – Lines 595-596), while the simulation was performed in water for 1 microsecond and had not reached an equilibrium. Also, TFE was not present in the simulation – we only performed the simulation in water.

-In first full paragraph on page 23 in Supplementary Information, we now state “We note that the circular dichroism measurements were made after incubating our protein for two hours with trifluoroethanol, during which conformational changes have significant time to proceed; this contrasts with the 1 microsecond simulation performed in water. Thus, a correspondence between the simulation result (amino acids mostly in a coil state) and the CD spectrum result (a significant fraction of amino acids in alpha helices) is not expected.”

8) Page 9. line 207.

Does Figure S9 mean Figure S8 (C)?

- We edited the figure numberings in supporting information. We thank the reviewer for the correction.

9) Page 10. line 208.

One of two conserved region mentioned here seem to be just a ligand binding site conserved among FABPs and FABP-like proteins. Therefore, again, I wonder why only SAHS proteins have a protecting ability.

-First, we have not tested FABPs and FABP-like proteins for protective ability, and to the best of our knowledge no one else has. Second, we imagine that the protective capability of SAHS proteins results partly from an ability to form a gel-like network that results from specific but undiscovered features of their sequences. We would not expect FABPs to have this capability because of a lack of natural selection for such a feature.

10) Page 12. Line 274.

The authors suggested that SAHS proteins do not bind a common ligand. What does “common ligand” mean here? To some extent, FABPs, FABP-like proteins, and other lipocalin proteins show ligand selectivity and bind different ligands. They do not have to bind a common ligand. That’s why they show amino acid sequence variations.

- A common ligand would be something like a fatty acid. We have now added a parenthetical note to this effect (line 355). The reviewer’s reasoning here is the same as ours.

Reviewer #2 (Remarks to the Author):

The manuscript (MS) by Lim et al. presents an experimental and computational study of secreted-abundant heat soluble (SAHS) proteins, one of the three types of proteins involved in the ability of tardigrades to survive in complete desiccation. The investigation of tardigrade proteins related to their capability to endure extreme stresses has aroused a great interest for their potential biotechnological and biomedical applications. This MS represents a novel contribution to this research by showing the role of SAHS proteins in protecting biomolecules from desiccation-induced damage and enhancing survival of desiccated bacterial cells.

The MS is well presented and well written. The experimental part investigating in vitro the protection effects of SAHS is correctly addressed and discussed. However, the computational part addressing with MD calculations an alleged structural change of SAHS proteins under desiccation suggested by CD spectra needs more elaboration. The protective mechanism associated with that structural change is “envisioned” (authors’s term) without a firm enough basis. My comments below address these impressions alongside other remarks.

- We appreciate the reviewer’s positive response.

-We have dialed back the discussion of the possible protective mechanism. We also eliminated use of the word ‘envision’.

In lines 55-56, the authors state that the disorder in TDPs is an “inference”. It should be pointed out that there currently are reliable predictors of protein disorder from sequence (doi.org/10.1038/s41592-021-01117-3). Structural disorder presented in the literature is in most cases directly predicted from sequence. In fact, for the three types of TDPs required for desiccation tolerance in the tardigrade *Hypsibius dujardini* presented in ref.15, disorder is directly based on the systematic prediction with IUPred (doi.org/10.1093/nar/gkab408) for the sequences of the 58 CAHS, 5 MAHS, and 42 SAHS studied there.

-We note that the authors of reference 15 (a seminal paper in this field), state on page 977 “These predictions strongly support previous evidence that CAHS and MAHS proteins are largely disordered, while by this approach SAHS proteins appear less disordered” and at the top of page 978 “These data, combined with heat solubility, circular dichroism, and bioinformatics approaches, show that many, if not all, tardigrade CAHS, and likely MAHS and SAHS proteins, are disordered”. The subsequent publication of the crystal structures of SAHS1 and SAHS4

would indicate that these are not disordered proteins. Reference 15 provided good experimental evidence that CAHS proteins were intrinsically disordered, but since the quoted statements about SAHS proteins were based only on computer-based predictions and subsequently shown to not be accurate, we want to err on the side of caution in interpreting such predictions.

-We consider an 'inference' to be essentially synonymous with 'prediction'. We now use the word 'prediction' in this section (line 61).

2. In lines 64-65 and then in lines 221-222, the authors state that describing SAHS proteins as intrinsically disordered proteins is not accurate. This is correct but it should be qualified by pointing out that whereas CAHS and MAHS are shown in ref. 15 to be completely or largely disordered, the 42 predictions for all the SAHS in the supplementary information of ref. 15 reveal that they are nearly completely ordered. Although this study refers to *H. dujardini*, its results could reasonably be assumed also valid for *Ramazzottius varieornatus*.

- Again, we note that the main text of reference 15 states "These data, combined with heat solubility, circular dichroism, and bioinformatics approaches, show that many, if not all, tardigrade CAHS, and likely MAHS and SAHS proteins, are disordered." The Supplementary Information Document S1 in reference 15 contains a large number of "Disorder tendency" plots with no figure legend or indication of how the plots are to be interpreted. The actual solved crystal structures of SAHS1 and SAHS4 are the best indication that the SAHS proteins are not disordered.

-We now note that, according to ref. 15, the SAHS proteins were predicted to be more ordered than CAHS or MAHS proteins. (lines 68-71)

3. In the paragraph introducing the structural features of SAHS proteins (lines 61-69), the authors refer to the crystal structures of RvSAHS1 and RvSAHS4. It should be also mentioned that there are predicted structures in the AlphaFold Protein Structure Database for all the SAHS proteins (12 from *R. varieornatus* and 1 from *H. dujardini*) studied in the MS. The comparison between crystal and AlphaFold structures of RvSAHS1 and RvSAHS4 demonstrates the high reliability of the predicted structures and provides a basis for the compared structural study mentioned in points 4 and 9 below.

- We consider that the AlphaFold structures are generated based on pre-existing solved structures, which would include at least the structure of SAHS1, which was published in 2017. Therefore the correlation between the structures predicted by AlphaFold and the solved structures may not be meaningful because they are not independently derived.

*-Also, the SAHS structures in the AlphaFold already-predicted dataset are monomers. Part of our choice in obtaining a new AlphaFold prediction of three different copies of SAHS1 was to see if protein-protein interactions might be revealed. As noted above in point 6 from Reviewer 1, AlphaFold did generate a novel prediction of an interaction between SAHS monomers, which we now describe in Supplementary Figure S10 and briefly mention in the Discussion (Lines 452-454). We also mention that AlphaFold structures of all of the SAHS proteins from *R. varieornatus* and *H. exemplaris* are available. (lines 118-119)*

4. Since there are 3D model structures for the 13 SAHS studied, it should be interesting to compare them with each other to detect differences that could be relevant to some of the issues addressed in the MS such as for instance, their (apparently) distinct subcellular localization or the different size of their internal cavity. While the authors address this issue by studying sequence (pages 4 and 5), it might be interesting to do it by studying structure. In this regard, they can check that in Dali (<http://ekhidna2.biocenter.helsinki.fi/dali/>) the tool “All against all” provides a correspondence analysis plot and a dendrogram in which the 13 SAHS are organized in different groups on the basis of the slight structural differences detected in their 13 AlphaFold models.

- In response to the reviewer's comment, we did use the Dali server to compare the available 3D structures of 10 SAHS proteins and 2 FABP proteins. Several SAHS proteins were excluded because either their AlphaFold structures were unavailable (RvSAHS8) or they were based on possibly incorrect sequence annotation (RvSAHS9,10,12). This analysis generated the dendrogram below. However, this is rather similar to a dendrogram generated by sequence alone and does not provide new insights or reflect on our conclusions, so we have not added it to the manuscript.

Structural comparison of SAHS proteins and relevant binding proteins. (A) Structural dendrogram showing distinctly grouped proteins. (B, C) RvSAHS1 structure showing (B) sequence and (C) structure conservation, where blue represents high conservation and red represents low conservation.

5. Given that there are crystal structures for RvSAHS1 and RvSAHS4, it should be interesting to compare the percentages of the secondary structure elements obtained from CD spectra in the 0% case in Figures 4B and S6 with that directly obtained from the crystal structures. That comparison might provide an assessment for the

quantitative considerations in lines 195-199. Although one could argue that the environmental conditions of proteins for X-ray diffraction and CD work are different, this also happens for CD measurements and MD simulations that are used in the MS to address the structural change of RvSAHS1.

- In the manuscript, we now state "The CD spectra for SAHS1 and SAHS4 at 0% in aqueous solution correlate well with the solved structures of these proteins, which have a single short alpha helix and are dominated by beta strands." Lines 230-232. This is a point worth making.

6. A filamentous, fibrous network structure is obtained in TEM images for RvSAHS1 dried at two distinct concentrations (Fig. S7). This interesting result is succinctly (lines 182-187) presented but it deserves further highlighting. An analogous finding for TDPs is addressed in ref. 4 (its very title announces it) but the authors do not mention it. Also published in 2022 is other article (not referenced in the MS) that deals with assembly of TDPs into fibrous gels in response to environmental stress (doi.org/10.1002/anie.202280161).

- As suggested, we now mention these points in the legend to Figure S7, and cite the new reference suggested by the reviewer in this section.

7. The MD simulation on RvSAHS1 should be considerably clarified. Not only their methodological details are unclear, but it should be explained how a MD simulation on a system prepared with the protein properly solvated is used to study the protein in desiccation conditions. Specifically, the following points should be addressed. (a) An initial structure is generated from sequence "...using alphaFold" (lines 506-507) but not details about the calculation are presented if this is done with AlphaFold2 software (which is not properly referenced). Given that this structure is available in the AlphaFold Database (which is not properly referenced, neither), why did the authors generate it? (b) No details about the solvation box other than the TIP3 water model is used (line 508). (c) If the solvated structure is equilibrated at 310 K for 1 ns, how is the thermal connection with the simulation at 550 K done? (d) No details about the ensemble in which the simulation is performed are given. (e) No details about the methods to control T and P constant (if the NPT ensemble is used) are given. (f) No details about the truncation of nonbonded pair VdW interactions and long-range electrostatic interactions are given. (g) It is stated (line 192) that the simulation is performed at 550 K "...to induce breaking of hydrogen bonds", yet at that temperature there is no liquid water. (h) No details about the method used to identify secondary structure along the simulation (Fig. 5B) are given. (i) If H-bonds are not present due to the high temperature, one cannot speak of "alpha helices" or "beta sheets". One should instead speak of (Phi, Psi) dihedral angles reminiscent of alpha helix or beta strand conformations. But then a method to compute those angles along the simulation must be presented.

-The reviewer makes a good point in that in the initial submission our description of the simulation was rather minimal. We break out these reviewers' comments below and respond to each. Key elements of these points are now covered in the new Supplementary Information section "Supplementary Computational Simulation Methods" (pages 22-25).

The MD simulation on RvSAHS1 should be considerably clarified. Not only their methodological details are unclear, but it should be explained how a MD simulation on a system prepared with the protein properly solvated is used to study the protein in desiccation conditions.

-We now clarify in this new section that the simulation does not seek to replication desiccation conditions, which would involve crowding with other proteins, but only protein denaturation, which is just one aspect of the process by which SAHS proteins might convert into a different state.

Specifically, the following points should be addressed.

(a) An initial structure is generated from sequence "...using alphaFold" (lines 506-507) but not details about the calculation are presented if this is done with AlphaFold2 software (which is not properly referenced).

- Alphafold2(Jumper, John, Richard Evans, Alexander Pritzel, Tim Green, Michael Figurnov, Olaf Ronneberger, Kathryn Tunyasuvunakool, et al. 2021. "Highly Accurate Protein Structure Prediction with AlphaFold." Nature, July. <https://doi.org/10.1038/s41586-021-03819-2>. (<https://github.com/google-deepmind/alphafold>). This is now cited in the new section (middle of page 23).

Given that this structure is available in the AlphaFold Database (which is not properly referenced, neither), why did the authors generate it?

- The structure available in the AlphaFold Database is a monomer. Initially, we generated AlphaFold-predicted structures for 1, 2, 3, 4, 8, and 16 molecules, and various putative structural complexes were revealed. The choice of using the three-member set for molecular dynamics simulations was to strike a balance between capturing potential protein-protein interactions while not building an inefficient simulation system for long timescales. Explicit solvent, including water, was incorporated in our simulation to replicate desiccation conditions better, recognizing that while desiccation removes water, other molecules remain present and provide a variety of inter-molecular interactions that influence intra-molecular interactions. This approach contrasts with past studies that simulated proteins in a vacuum, a method noted by the reviewer as not physiologically relevant. Desiccation subjects proteins to "extreme conditions," including fluctuations in pressure, salt concentrations, and mechanical forces. While replicating all these conditions is beyond the scope of this study, we have attempted to mimic such extreme conditions by employing high temperatures while maintaining pressure. At these elevated temperatures, hydrogen bonds break or break and reform quickly and the protein's intramolecular structure is disrupted, yet water molecules continue to interact with the protein.

b) No details about the solvation box other than the TIP3 water model is used (line 508).

- In the new section (middle of page 24, Supplementary Information), we now state: "A 100 Å simulation box with TIP3 water as solvent was generated using Ambergtools and the amber forcefield (ff14SB) (<https://ambermd.org/doc12/AmberTools13.pdf>; <https://ambermd.org/Manuals.php>). Na⁺ and Cl⁻ ions were added to achieve a neutral charge. Simulations were performed using Openmm 7 (<http://docs.openmm.org/7.7.0/developerguide/>),

*periodic boundary conditions, and a Particle-Mesh Ewald with a cutoff of 1*nanometers and an Ewald error tolerance of 0.0005. A Monte Carlo barostat was used at 1-atmosphere pressure and an interval of 25 using the Langevin Integrator. Following equilibration at 310°K, the simulation was extended from a restart checkpoint, and velocities reset to the 550°K temperature.”*

(c) If the solvated structure is equilibrated at 310 K for 1 ns, how is the thermal connection with the simulation at 550 K done?

- See last sentence of the preceding paragraph.

(d) No details about the ensemble in which the simulation is performed are given.

-The ensemble was generated from the homotrimer structure predicted by AlphaFold2 and included in the solvation box as described above.

(e) No details about the methods to control T and P constant (if the NPT ensemble is used) are given.

-As described above, A Monte Carlo barostat was used at 1-atmosphere pressure and an interval of 25 using the Langevin Integrator.

(f) No details about the truncation of nonbonded pair VdW interactions and long-range electrostatic interactions are given.

*-Described above - Periodic boundary conditions and a Particle-Mesh Ewald with a cutoff of 1*nanometers and an Ewald error tolerance 0.0005.*

(g) It is stated (line 192) that the simulation is performed at 550 K “...to induce breaking of hydrogen bonds”, yet at that temperature, there is no liquid water.

-If one were to instantaneously heat a protein solution from 310°K to 550°K, the water would not boil away in one microsecond, the timescale of our simulation. The water would simply be superheated. Using artificially high temperatures in simulations is a common trick for accelerating rates and looking at events that might occur on a timescale that is not practical to model.

-In the physiological context, where water is removed by desiccation, the protein is not in a vacuum – rather the concentration of macromolecules is increased. The rationale for using an elevated temperature is not to mimic all aspects of desiccation but rather to mimic possible denaturing conditions experienced under desiccation. These conditions experienced in a proteinaceous milieu may include increased salt concentration due to water removal, and also mechanical forces. To mimic all of these conditions is not in the scope of this study.

(h) No details about the method used to identify secondary structure along the simulation (Fig. 5B) are given.

-The Python package MDTraj was used. Specifically, the protein secondary structure (DSSP) secondary structure assignments function (https://mdtraj.org/1.9.4/api/generated/mdtraj.compute_dssp.html). This function implements the assignment based on the following reference -- Kabsch W, Sander C (1983). “Dictionary of

protein secondary structure: pattern recognition of hydrogen-bonded and geometrical features. *Biopolymers* 22 (12): 2577-637. doi:10.1002/bip.360221211

-The simplified version was executed, which groups the secondary structure into helical, strand, and coil. "Helical" includes Alpha helix, 3-helix (3/10 helix), and 5 helix (pi helix). "Strand" includes residues in isolated beta-bridge and extended strand, participating in beta ladder. The "Coil" includes hydrogen-bonded turns and bends.

(i) If H-bonds are not present due to the high temperature, one cannot speak of "alpha-helices" or "beta sheets". One should instead speak of (Phi, Psi) dihedral angles reminiscent of alpha helix or beta strand conformations. But then a method to compute those angles along the simulation must be presented.

-Secondary structure in the context of these MD simulations refers to structure that exhibit the same dihedral angles as conventional secondary structure descriptors. Therefore, we utilize the secondary structure descriptors. The method to compute is described in the previous point (Kabsch W, Sander C (1983)). Hydrogen bonds are observed in the high-temperature simulation, but simply don't last very long.

8. Irrespective of the concerns raised in the preceding point, the main effect seen in Fig. 5B is the change of beta sheet to coil: orange and red plots are nearly specular images. Compared to this, the slight increase shown by the blue plot (alpha helix) is a secondary effect. Moreover, the structure at 100 ns in Fig. 5C shows a clearly helical arrangement only in chain A (orange) while chains B and C exhibit irregular conformations without a clear helical pattern. In view of these results, the claims that "...SAHS proteins might undergo a dramatic transition from a beta sheet conformation to a predominantly alpha-helical state" (lines 212-213) or that (SAHS proteins) "...reconfigure into a completely different conformation dominated by alpha-helices" (lines 279-280) or "...shift from a primarily beta-sheet structure to an alpha-helical structure" (lines 293-294) seem slightly exaggerated. In this regard, the conjectural statements about SAHS proteins acting as deformable packing material (lines 230-236) speak much more appropriately of "disrupting the beta sheet structure" (line 234).

-The reviewer makes a good general point that our statements about conversion to an alpha helical structure are exaggerated, given that the data supporting a specific conversion to alpha helices (as opposed to some undefined conformation) are limited. We have now modified these statements as follows:

*-"Based on this analysis and data presented here, we propose a model by which the thirteen SAHS proteins encoded by *R. varieornatus* undergo a transition upon dehydration in which water molecules equilibrate out of the cavity and crowding and pressure of other proteins then disrupts the structure of the SAHS proteins, which might then reconfigure into a completely different conformation." (Line 356-360.)*

-"Based on CD analysis, the SAHS proteins appear to undergo a shift from a primarily beta-sheet structure to an alpha-helical structure in increasing TFE concentrations (thought to be desiccation-mimicking conditions), and a 1 microsecond molecular dynamic simulation showed that short alpha-helices can transiently form upon denaturation of proteins." (Line 389-393)

-We also deleted from the Discussion a particularly speculative paragraph (formerly between lines 449-450) that focused on alpha helices.

9. The discussion on the organization of amino acids with respect to the cavity and its entry/exit point compared with FABPs is interesting and provides information useful to unveil the possible movement of water in the changes occurring under desiccation. The results of the comparison provided by Dali mentioned in point 4 above showing slight differences in the size of the cavity in some of the 13 SAHS proteins could be probably relevant here.

-We consider that simple visual inspection of the cavities in models of the different SAHS proteins would be more directly informative. Also, the hydrophobicity/hydrophilicity of the amino acid side chains would be relevant to the ability of water to escape. Nonetheless, we simply wanted to bring it to the reader's attention that the N-linked glycosylation sites are clustered around the opening to the cavity, which may affect the entry/exit of water. At this point we do not have a prediction about how the various SAHS proteins might differ in this regard.

10. In lines 305-314, the authors highlight *R. varieornatus* as compared with *H. exemplaris* with regard to the large number of SAHS proteins wondering why the former tardigrade encodes as large SAHS members as 13. **They should remark that, as mentioned in points 1 and 2 above, ref. 15 presents 42 SAHS members in *H. dujardini*.**

*- We acknowledge that our original wording was rather misleading as it implied that only *R. varieornatus* encodes many SAHS variants. We make this point in lines 68-71. However, the 42 SAHS proteins described in reference 15 are from several different species, not just *H. exemplaris/dujardini*. We also have changed this wording, and now mention that *H. exemplaris* also encodes multiple SAHS variants based on ref. 15. (lines 403-405)*

11. Incidentally, terms *Hypsibius dujardini* and *Hypsibius exemplaris* are frequently used in the literature as synonymous and this MS also does it. But strictly they are not: see doi.org/10.11646/zootaxa.4415.1.2.

*-According to the reference cited by the reviewer and also according to Genbank, the sequence we refer to as HeSAHS4 is in fact from *H. exemplaris*. The source strain is referred to as *H. dujardini* in reference 15 but was subsequently reclassified as *H. exemplaris* according to the reference cited by the reviewer. We have eliminated mention of *H. dujardini* in the main text, and fixed the mention of *H. dujardini* in Table S1.*

The Discussion presents a valuable synthesis of the results on RvSAHS proteins reported in the MS. The interesting findings that two bacteria could be stabilized against desiccation by a single SAHS protein is properly emphasized and a novel mechanism for the protective action of SAHS proteins is also tentatively proposed. Except for the recurrent claims that SAHS proteins adopt alpha-helical structures when the evidence presented rather suggests that beta-sheet structures are just unfolded, the model envisioned (their term) for that mechanism together with the experimental results make this MS an interesting and original contribution. Provided that the concerns raised in this

review are properly addressed, this work could be relevant for researchers of tardigrade proteins involved in the outstanding abilities of these animals to survive a wide range of environmental stresses.

-We thank the reviewer for the encouraging response and the useful comments – all of which have been addressed.

REVIEWERS' COMMENTS:

Reviewer #1 (Remarks to the Author):

The authors have addressed all of my points in my review.

Reviewer #2 (Remarks to the Author):

In this revised version of the manuscript by Lim et al., the authors have addressed satisfactorily all the concerns raised in my previous review. In its current form, I recommend publication.